# Research on Quality Decisions and Coordination with Reference Effect in Dual-Channel Supply Chain

**Zhou Xideng [1,2,3], Xu Bing [2,*], Xie Fei [1] and Li Yu [4]**

[1] School of Business Administration, Nanchang Institute of Technology, Nanchang 330099, China; chowxd99@nit.edu.cn (Z.X.); xf514@nit.edu.cn (X.F.)
[2] School of Management, Nanchang University, Nanchang 330031, China
[3] Soft Science Research Base of Water Security and Sustainable Development in Jiangxi Province, Nanchang 330099, China
[4] School of Public Finance and Public Administration, Jiangxi University of Finance and Economics, Nanchang 330013, China; xiana520@stu.jxufe.cn
[*] Correspondence: xubing99@ncu.edu.cn; Tel.: +86-0791-8396-8410

**Abstract:** Although supply quality management has been studied extensively, one important marketing phenomenon, that is, reference effect has been rarely considered in dual-channel supply chain quality management literatures. In fact, the quality reference effect is also an important factor which influences consumer purchasing behavior. We aim to explore the influence of the reference effect on the optimal decisions and performance of a dual-channel supply. Thus, we formulate dynamic models that include the product quality reference effect and the service quality reference effect in a dual-channel supply chain system consisting of a manufacturer and a retailer under the different decision-making scenarios. Utilizing differential game theory, optimal decisions are obtained for the product quality and service quality decision under the different decision-making scenarios. In addition, the optimal decisions and profits are compared, then a service cost-sharing coordinating mechanism is proposed and proven to be effective in the supply chain system. The main results show when the initial reference service quality is low, the consumer service quality reference effect is beneficial to the manufacturer. The spillover effect of service quality is not conducive to the retailer and the manufacturer. When the initial reference product quality is low, both online and offline product quality reference effects are beneficial to the retailer and the manufacturer. The stable (or final) reference quality will not be affected by the initial reference quality. The sum of the two members' profits under decentralized decision making is less than the total profit of the supply chain under centralized decision making. We design a cost-sharing coordinating mechanism to eliminate the double marginal effect.

**Keywords:** dual-channel; quality; reference effect; differential game; coordination

## 1. Introduction

Under the new mode of economic development, economic growth has changed from mainly relying on increasing the consumption of material resources to mainly relying on scientific and technological progress, improving the quality of workers and management innovation. There is no doubt that we need to increase the development of circular economy. A circular economy can be defined as an economic model aimed at the efficient use of resources through waste minimization, long-term value retention, reduction of primary resources, and closed loops of products, product parts, and materials within the boundaries of environmental protection and socioeconomic benefits [1]. The objective of circular economy is to extract the advantage of materials, enemy and wastes of

an industry [2]**.** Some scholars have carried out relevant research in the context of ecological economy [3,4].

The development of circular economy must improve the quality of products and service, because without quality support, circular economy cannot adhere to the principle of "reduction, reuse, recycling", and cannot continue to develop. The best way to save is to improve product quality for extending product life, and improve service quality for promoting economic operation efficiency. Quality is the key content of social wealth, the important driving force of a country's economic and social development and the main evaluation scale of the level of economic and social development. In February 2019, all Dyson's products were removed from its list of recommended vacuum cleaners, according to *consumer reports*, the leading US consumer magazine. The reason was that consumers cannot expect these vacuum cleaners to last. According to the survey data, nearly half of Dyson's hand-held vacuum cleaners will be damaged within five years, with the cracking rate higher than any other brands [5]. In 2014, Volkswagen Sagitar took recall and maintenance measures due to the product quality problem of rear axle longitudinal arm fracture, but still could not eliminate the safety risk. Product quality problems not only reduce the reputation of enterprises, but also waste resources [6]. Therefore, many enterprises take more measures to increase investment in product quality and service quality. Yulai automobile always attach great importance to product and service quality. According to relevant statistics, from 2016 to the first half of 2019, the R & D cost of Yulai is up to 10.445 billion RMB, the amount spent on sales and management costs is 11.571 billion RMB, and the amount spent on operating expenses is 22.016 billion RMB [7]. Due to the fierce market competition, quality has always been the cornerstone of the survival and development for a enterprise. Therefore, how to design the optimal product quality and service quality is a very important matter, which is the main content of this paper.

In recent years, the public has been paying greater attention to product quality. In specific segments of the market in some industries, competition is shifting from price to quality [8,9]. Therefore, more firms are adopting quality improvement as a powerful competitive tool in the market. And in supply chain management, how to design optimal product quality and service quality has attracted extensive attention. Another important marketing phenomenon, the reference effect, has also been studied. However, most scholars focus on the price reference effect and seldom consider the quality reference effect. According to Fibich et al. [10], the reference price is defined as the weighted average value of the product price observed in previous purchase experience. The further away from the present, the less the weight of the price corresponds to the past moment. The influence of the asymmetric reference price effect on the price strategy of the enterprise is analyzed in infinite and limited time lengths, respectively. In addition, some researchers have extended the concept of reference point to product quality [11–14]. According to Fibich et al. [10] and He et al. [13], we can define reference quality as the quality $r$ that consumers have in mind and to which they compare the current quality $z$ of a product. When $r > z$, consumers will form a sense of "gain" that will increase demand; otherwise, they will form a sense of "loss". The above research works are only related to the product quality reference effect. In the paper, we further divide the quality reference effect into product quality reference effect and service quality reference effect. In particular, in their frequent traditional shopping experience previously, consumers have formed an expected service quality (called the reference point). When the actual service quality provided by the retailer is higher than the expectation of consumers, consumers will form a sense of gain and be attracted to shopping once again. Since the reference effect can influence consumers' purchase behavior, it is meaningful for enterprises to explore how reference effect affects quality decision making and profit. Through the research of this paper, it can provide guidance for enterprises in the decision making process.

With the growing maturity of e-commerce technology, increasing numbers of manufacturers have opened online channels on the basis of the original traditional retailer channels. The development of a manufacturer's network channel will erode the retailer's market demand and lead to channel competition and conflict. Facing threats of the invasion of manufacturers' direct marketing channels to the traditional offline market, retailers begin to provide service—that online stores are not able to provide (such as a shopping environment, touchable goods, enthusiastic shopping guide

service and so on)—to help consumers to better understand the product performance and promote demand. Therefore, in the competitive dual channel supply chain, one of the main objectives of this paper is to investigate the impact of the service quality reference effect on all members.

The research objectives of this paper are as follows: First, in the context of circular economy, considering the influence of the reference effect on consumers' purchase decision making, effective strategies are developed for enterprises. Second, according to the degree of cooperation between the members, this paper compares the profits of each member in different decision making scenarios, then provides a win-win coordination mechanism for the sustainable development of the enterprise alliance. Third, to clarify the impact of reference effect on the profits and decisions in dual channel supply chain is helpful for managers to adjust strategies flexibly in different scenarios.

This paper proceeds as follows. A literature review is presented in Section 2. Section 3 describes the notations and proposes basic assumptions and three models under the three scenarios are proposed, and solutions are derived in the following sections. The optimal decisions and profits under the three scenarios are compared and analyzed in Section 4. A coordinating mechanism is proposed in Section 5. Section 6 provides a numerical analysis. Finally, Section 7 provides the concluding remarks.

## 2. Literature Review

This paper involves research issues including dual-channel supply chain and quality.

In recent years, with the increasing development of e-commerce technology and with the change of the way consumers shopping, more and more firms have opened online channels on the basis of the original traditional retailer channels. Dual-channel supply chain has become an emerging field of supply chain management research. Chiang et al. [15] found that a manufacturer can reduce the channel competition by reducing the wholesale price after introducing a direct marketing channel. Considering that consumers are either brand loyal or retailer loyal, Kumar and Ruan [16] inferred the conditions under which manufacturers can benefit from additional direct channels. Cai et al. [17] studied the influence of the pricing mechanism on channel competition in the dual channel supply chain and found that a uniform pricing strategy can alleviate channel conflict. Ryan et al. [18] found that when the retailer's market share is larger than the manufacturer's direct channel market share, the manufacturer will implement a price discrimination strategy. Yan et al. [19] found that a manufacturer can encourage the retailer to improve their service level by introducing online channels. Lu and Liu [20] studied the impact of opening up network channels on the profitability and behavior of manufacturers and physical retailers in the supply chain. Li et al. [21] studied the impact of service provided by retailers in a dual channel supply chain composed of a manufacturer and a retailer with fair concern. It was found that whether the retail service quality is beneficial to both the manufacturer and the retailer depends on the loyalty of the consumer to the traditional retail channel. He et al. [14] investigated how firms should incorporate the reference quality effect under different business models: a pure offline store case, a pure online store case, and an O2O(online-to-offline)case. Ni et al. [22] examined a two-echelon supply chain with an upstream supplier and a downstream manufacturer transacting an intermediate product via direct bilateral contracting and futures market channels with differentiated productivities. Li et al. [23] investigated the strategic effect of return policies in a dual-channel supply chain. Zhang et al. [24] integrated the product quality and returns caused by quality problems into the design of a dual-channel coordination mechanism in closed-loop supply chains. Modak and Kelle [25] considered to develop a dual-channel supply chain strategy under price and delivery-time dependent stochastic customer demand. Alizadeh-Basban and Taleizadeh [26] studied dual channel green supply chain considering sale's efforts, delivery time, and hybrid remanufacturing. The delivery time in the online channel as well as the distributor sales effort in the retail channel were devised. The above literature studied the pricing [15,17,18,25], service level [19,21], delivery [25,26], return [23,24], remanufacturing [26], contract model [22] and consumer behavior [14,21] of dual channel supply chain. However, there are few works on product quality and service quality in dual channel supply chain, as well as the effect of service spillover on channel members.

In recent years, the news reports on product quality have not been rare, which has also attracted the attention to supply chain quality. Boyaci and Gallego [27] considered two competitive supply chains with a supplier and a retailer and studied three kinds of competition scenarios between supply chains. Gurnani and Erkoc [28] constructed the demand function of product quality and marketing effort and compared three different supply chain contract forms. Chenavaz and Regis [29] and He et al. [14] studied the dynamic quality decision making of enterprises in which consumers have reference quality behavior. Zhu et al. [30] took a three-level supply chain as the research object, constructed three distribution channel models, and compared the quality strategies in different channel models. Xiao et al. [31] studied four kinds of supply chain decision making scenarios and proposed that supply chain cooperation could not necessarily improve the product quality of the supply chain. From an irrational perspective, Liu et al. [32] considered that the downstream retailers have a preference for loss aversion and studied the supply chain quality decision making in a centralized and decentralized situation. The above works studied a case in which the manufacturer or supplier designs the product quality to promote the market demand, but did not consider that the retailer improves service quality.

The research on service quality has also attracted the attention of many scholars. Tsay and Agrawal [33] studied the dynamic model of service competition with capacity constraints. Shirui Xia et al. [34], Ali et al. [35], Zhou et al. [36]and other scholars have also studied the service quality in the supply chain. In the dual channel supply chain model, Pu [37] considered the offline service quality spillover effect on the online channel and designed a cost-sharing contract to achieve supply chain coordination. Qin et al. [38] studied the competition and cooperation strategies between two competitive logistics service providers and the online retailer. Ren et al. [39] and Dan et al. [40] both studied service competition online and traditional channels.

Improving product quality is the foundation for survival and development for an enterprise, and how to effectively transferring products to the consumer, to a certain extent, also needs effective service quality, both of which are very important for an enterprise to improve their performance and enhance competitiveness. Therefore, it is necessary to study the coordination of the supply chain considering both product quality and service quality. To date, most of the literature has focused on the unilateral research of product quality or service quality. A few scholars have assumed that market demand is affected by both product quality and service quality and studied supply chain coordination [24]. However, in the case of the dual channel supply chain, there are few works regarding the combination of product quality and service quality strategy.

Our paper proposes the following major differences from the aforementioned literatures. Firstly, the influence of the product quality reference effect and the service quality reference effect on demand is considered. Secondly, from the perspective of reference effect, we explore and analyze the impact of service quality spillover on online and offline channels.

On the basis of the above literature, this paper further expands the scope of the research to consider the consumer behavior (called the product quality reference effect and the service quality reference effect) in the dual channel supply chain, formulating a Nerlove-Arrow model. Utilizing differential game theory, optimal decisions are obtained for the product quality, service quality decision-making and coordination mechanism.

## 3. Model Development

Consider a supply chain where a manufacturer sells a final product through both a traditional channel retailer and their own online channel in parallel: the manufacturer invests in product quality improvement, where $z(t)$ denotes the product quality level over time $t$, while the retailer invests in the service quality improvement, where $f(t)$ denotes the service quality level over time $t$. See Table 1 for the notations and related descriptions.

**Table 1.** Notation and description.

| Notation | Description |
| --- | --- |
| $f(t)$ | Retailer's service quality at time $t$, which is a decision variable. |

| | |
|---|---|
| $z(t)$ | Manufacturer's product quality at time $t$, which is a decision variable. |
| $\theta_{ft}$ | Service quality's effectiveness on market demand in traditional channel. |
| $\theta_{zt}$ | Product quality's effectiveness on market demand in traditional channel. |
| $\theta_{ze}$ | Product quality's effectiveness on market demand in online channel. |
| $\rho_1$ | Retailer's marginal profit in traditional channel. |
| $\rho_2$ | Manufacturer's marginal profit in traditional channel. |
| $\rho_3$ | Manufacturer's marginal profit in online channel. |
| $k_1$ | Cost parameter associated with service quality improvement by the retailer. |
| $k_2$ | Cost parameter associated with product quality improvement by the manufacturer. |
| $\lambda$ | Discount rate. $\lambda > 0$. |
| $\phi^d$ | Cost-sharing rate provided under decentralized decision making. |
| $\phi^{cs}$ | Cost-sharing rate provided under coordinating mechanism. |

It is assumed that consumers' expectations (called the reference point) with respect to product quality and service quality evolve over time and are based on their frequent shopping experience previously with the product. When the quality level is higher than the reference point, consumers will form a sense of "gain"; otherwise, they will form a sense of "loss".

In our study, which is based on the reference effect formation of Kopalle and Winer [41] the changes of the reference service quality and product quality are given, respectively, by Equations (1) and (2):

$$\dot{r}_f(t) = \alpha_1 (f(t) - r_f(t))$$
$$r_f(0) = r_{f0} \tag{1}$$

$$\dot{r}_z(t) = \alpha_2 (z(t) - r_z(t))$$
$$r_z(0) = r_{z0} \tag{2}$$

where $r_f(t)$ is the consumer's reference service quality level over time $t$, $f(t)$ is the retailer's actual service quality level, $r_z(t)$ is the consumer's reference product quality level over time $t$, and $z(t)$ is the manufacturer's actual product quality level. $\alpha_1$ and $\alpha_2$ indicate service quality memory parameter and product quality memory parameter, respectively, $\alpha_1 > 0$, $\alpha_2 > 0$.

Compared with online shopping, consumers are more likely to be affected by the service level of traditional retailers. Traditional retailers can provide a service that online stores are unable to provide, such as a shopping environment, touchable goods, enthusiastic shopping guide service and so on. In order to highlight this difference, this paper only considers the traditional offline retailers' service quality without considering manufacturers' online service quality. Based on the above assumptions, the traditional channel demand $d_t$ is affected by product quality, reference product quality, service quality and reference service quality. The online channel demand $d_e$ is affected by product quality, reference product quality, reference service quality of the traditional channel. The demand functions in the traditional channel and online channel are as follows:

$$d_t(t) = a_t + \theta_{ft} f(t) + \beta_{ft}[f(t) - r_f(t)] + \theta_{zt} z(t) + \beta_{zt}[z(t) - r_z(t)]$$
$$d_e(t) = a_e + \theta_{ze} z(t) + \beta_{ze}[z(t) - r_z(t)] + \beta_{fe}[f(t) - r_f(t)] \tag{3}$$

where $a_t$, $a_e$, $\theta_{ft}$, $\theta_{zt}$, $\theta_{ze}$, $\beta_{ft}$, $\beta_{zt}$, $\beta_{ze}$ and $\beta_{fe}$ are positive constants. $a_t$ and $a_e$ indicate potential sales volume in traditional channel and online channel respectively, $a_t > 0$, $a_e > 0$. The expression $\theta_{ft} f(t)$ represents the actual service quality effect on traditional demand. The expression $\beta_{ft}[f(t) - r_f(t)]$ represents the reference service quality effect on traditional channel demand. The expression $\beta_{fe}[f(t) - r_f(t)]$ represents the spillover effect of service quality in traditional channel on online demand. The expression $\theta_{zt} z(t)$ and $\theta_{ze} z(t)$ represent the actual product quality effect on traditional and online demand respectively. The expressions $\beta_{zt}[z(t) - r_z(t)]$ and $\beta_{ze}[z(t) - r_z(t)]$ represent the

reference product quality effect on traditional and online demand, respectively. The relationship between related variables and demand in this article is shown in Figure 1.

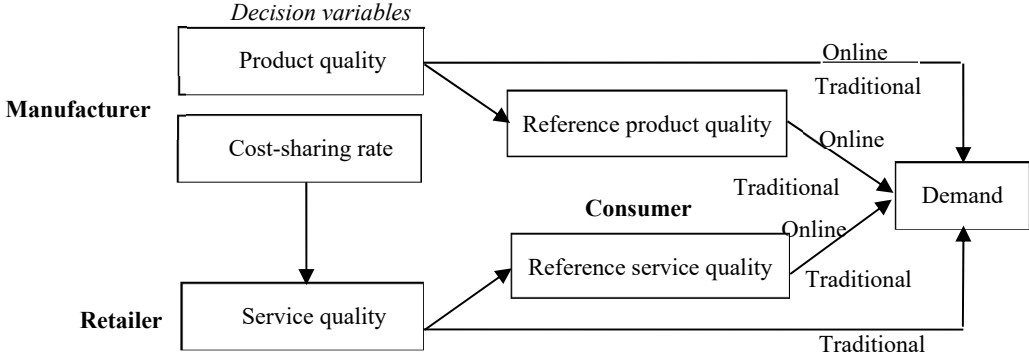

**Figure 1.** Relationships between related variables and demand.

In our model, we assume that the marginal profits $\rho_1$ , $\rho_2$ , $\rho_3$ are kept as constants for the following reason. The main purpose of this paper is to investigate the influence of the reference effects on the quality of channel members' decisions and explore an effective contract that could coordinate the supply chain.

Similar to previous studies [12,41], the product and service quality cost functions are quadratic with respect to the quality level, as specified by

$$c_r = \frac{1}{2}k_1 f^2(t)$$
$$c_m = \frac{1}{2}k_2 z^2(t)$$

(4)

where the manufacturer's product quality improvement cost $c_r$ and the retailer's service quality improvement cost $c_m$ are the increasing functions of $z(t)$ and $f(t)$, respectively, which meet the conditions of $c_z''(z) > 0$ , $c_r''(f) > 0$ .

According to the extent of cooperation between enterprises, in general there are two possible relationships between enterprises, namely, decentralized decision making without a cost-sharing contract (called model D) and decentralized decision making with a cost-sharing contract (called model DC). Then this paper puts forward the centralized decision (called model C), and compares the total profit of supply chain in three decision making scenarios. Finally, a win-win solution is proposed, which is the model CS. It also proves the importance of cooperation between enterprises.

We start by analyzing the first scenario in which each member makes decisions without a cost-sharing program (model D). In this scenario, the manufacturer does not share the service quality improvement cost with the retailer and all members make independent decisions. We use the superscript '*n*' to signify this scenario. In the second scenario, each member makes decisions within a cost-sharing construct, in which the manufacturer promises to share the service quality improvement cost of the retailer, and all members still make independent decisions (model DC). We use the superscript '*d*' to signify this scenario. In the third scenario, the manufacturer vertically integrates with the retailer. That is to say, the manufacturer and the retailer are regarded as two departments in an enterprise, which maximizes the profit of the enterprise by designing the optimal service quality and product quality (model C). We use the superscript '*c*' to signify this scenario. In the fourth scenario, the supply chain is coordinated by introducing a cost-sharing coordination mechanism. Under the scenario, the total profit of the centralized decision making supply chain is reasonably distributed between the manufacturer and the retailer through the cost sharing rate, while both the manufacturer and the retailer adopt the optimal strategy of the centralized decision making (model CS). We use the superscript '*cs*' to signify this scenario.

*3.1. Model D*

In this scenario, the manufacturer and the retailer make decisions respectively without a cost-sharing program. Each member tries to maximize the present values of its profit. All members carry out Stackelberg's differential game, where the manufacturer plays as the dominant while retailer as the follower in the infinite planning period. By serving as a benchmark model, this part aims to compare with the following models in order to propose the supply chain coordination mechanism.

The profit functions for two firms are represented by the following equations:

$$\pi_r(t) = \rho_1 d_t(t) - \frac{1}{2} k_1 f^2(t)$$

$$\pi_m(t) = \rho_2 d_t(t) + \rho_3 d_e(t) - \frac{1}{2} k_2 z^2(t)$$

(5)

Note that the profits for the two firms change along with time *t*. Each member tries to maximize the present values of its profit in infinite time zone. With a common discount rate $\lambda > 0$, we have the following equations for the retailer

$$J_r^n = \int_0^\infty e^{-\lambda t} [\rho_1 d_t(t) - \frac{1}{2} k_1 f^2(t)] dt$$

(6)

For the manufacturer

$$J_m^n = \int_0^\infty e^{-\lambda t} [\rho_2 d_t(t) + \rho_3 d_e(t) - \frac{1}{2} k_2 z^2(t)] dt$$

(7)

**Proposition 1.** *Under decentralized decision making without a cost-sharing contract, the manufacturer's optimal product quality level and the retailer's optimal service quality level, respectively, are as follows:*

$$f^{n*} = \frac{\rho_1(\theta_{ft} + \beta_{ft})(\lambda + \alpha_1) - \alpha_1 \rho_1 \beta_{ft}}{(\lambda + \alpha_1)k_1}$$

$$z^{n*} = \frac{(\alpha_2 + \lambda)[\rho_2(\theta_{zt} + \beta_{zt}) + \rho_3(\theta_{ze} + \beta_{ze})] - \alpha_2(\rho_2 \beta_{zt} + \rho_3 \beta_{ze})}{(\lambda + \alpha_2)k_2}$$

(8)

The reference service quality path is given by

$$r_f^{n*}(t) = (r_{f0} - f^{n*})e^{-a_1 t} + f^{n*}$$

(9)

The reference product quality path is given by

$$r_z^{n*}(t) = (r_{z0} - z^{n*})e^{-a_2 t} + z^{n*}$$

(10)

Proof (See Appendix A).

Proposition 1 gives the manufacturer optimal product quality and the retailer optimal service quality under decentralized decision making without a cost-sharing contract. It can be seen that the service quality of retailers is only affected by the marginal profit $\rho_1$ instead of the marginal profit $\rho_2$, $\rho_3$. According to the optimal quality, it is easy to deduce the path of reference quality changing with time. By substituting Proposition 1 into Equation (6) and Equation (7), the profit function of the retailer and the manufacturer can be simplified as follows:

$$J_r^{n*} = \frac{1}{\lambda} \rho_1(a_t + \theta_{ft} f^{n*} + \theta_{zt} z^{n*}) - \frac{1}{\lambda + a_1} \rho_1 \beta_{ft}(r_{f0} - f^{n*}) - \frac{1}{\lambda + a_2} \rho_1 \beta_{zt}(r_{z0} - z^{n*}) - \frac{1}{2\lambda} k_1 f^{n*2}$$

(11)

$$J_m^{n*} = \frac{1}{\lambda}\rho_2(a_t + \theta_{ft}f^{n*} + \theta_{zt}z^{n*}) - \frac{1}{\lambda + a_1}\rho_2\beta_{ft}(r_{f0} - f^{n*}) - \frac{1}{\lambda + a_2}\rho_2\beta_{zt}(r_{z0} - z^{n*})$$

$$- \frac{1}{2\lambda}k_2z^{n*2} + \frac{1}{\lambda}\rho_3(a_e + \theta_{ze}z^{n*}) - \frac{1}{\lambda + a_2}\rho_3\beta_{ze}(r_{z0} - z^{n*}) - \frac{1}{\lambda + a_1}\rho_3\beta_{fe}(r_{f0} - f^{n*})$$

$$- \frac{1}{2\lambda}k_2z^{n*2} + \frac{1}{\lambda}\rho_3(a_e + \theta_{ze}z^{n*}) - \frac{1}{\lambda + a_2}\rho_3\beta_{ze}(r_{z0} - z^{n*}) - \frac{1}{\lambda + a_1}\rho_3\beta_{fe}(r_{f0} - f^{n*})$$

### 3.2. Model DC

In the supply chain structure dominated by the manufacturer, in order to encourage the retailer to improve service quality and promote sales, the manufacturer actively shares a certain rate $\phi^d$ of the retailer service quality improvement cost. Decision sequences of the two members are described as follows: (i) the manufacturer offers the cost-sharing rate for the retailer's service quality improvement cost and (ii) after observing the manufacturer' a cost-sharing contract, the retailer and manufacturer determine their quality improvement efforts along time $t$ simultaneously. We keep the cost-sharing rate fixed and calculate the quality improvement effort of the retailer and manufacturer utilizing differential game theory. Then we calculate the optimal cost-sharing rate.

Similarly, the profits for the value functions of the manufacturer and the retailer are obtained easily as follows

$$J_r^d = \int_0^\infty e^{-\lambda t}[\rho_1 d_t(t) - \frac{1}{2}(1-\phi^d)k_1 f^2(t)]dt \tag{12}$$

$$J_m^d = \int_0^\infty e^{-\lambda t}[\rho_2 d_t(t) + \rho_3 d_e(t) - \frac{1}{2}\phi^d k_1 f^2(t) - \frac{1}{2}k_2 z^2(t)]dt \tag{13}$$

**Proposition 2.** *Under decentralized decision making with a cost-sharing contract, the manufacturer's optimal product quality level and the retailer's optimal service quality level, respectively, are as follows:*

$$f^{d*} = \frac{\rho_1(\theta_{ft} + \beta_{ft})(\lambda + \alpha_1) - \rho_1\beta_{ft}\alpha_1}{(1-\phi^d)(\lambda + \alpha_1)k_1}$$

$$z^{d*} = \frac{\{\rho_2[(\alpha_2 + \lambda)\theta_{zt} + \lambda\beta_{zt}] + \rho_3[(\alpha_2 + \lambda)\theta_{ze} + \lambda\beta_{ze}]\}}{(\lambda + \alpha_2)k_2} \tag{14}$$

The reference service quality path is given by

$$r_f^{d*} = (r_{f0} - f^{d*})e^{-a_1 t} + f^{d*} \tag{15}$$

The reference product quality path is given by

$$r_z^{d*} = (r_{z0} - z^{d*})e^{-a_2 t} + z^{d*} \tag{16}$$

The proof of Proposition 2 is similar to the proof of Proposition 1 and is omitted.

**Proposition 3.** *Under decentralized decision making with a cost-sharing contract, the optimal service cost-sharing rate of the manufacturer is*

$$\phi^{d*} = \begin{cases} \dfrac{2B-A}{2B+A}, & 2B > A \\ 0, & 2B \leq A \end{cases} \tag{17}$$

where $B = \dfrac{\rho_2[\theta_{ft}(\lambda + \alpha_1) + \lambda\beta_{fe}] + \rho_3\lambda\beta_{fe}}{(\lambda + \alpha_1)}$, $A = \dfrac{\rho_1(\theta_{ft}\lambda + \beta_{ft}\lambda + \theta_{ft}\alpha_1)}{(\lambda + \alpha_1)}$.

Proof (See Appendix B): Put the above conclusions into Equation (12) and Equation (13), and simplify the profit functions of the retailer and the manufacturer as follows:

$$J_r^{d*} = \frac{1}{\lambda}\rho_1(a_t + \theta_{ft}f^{d*} + \theta_{zt}z^{d*}) - \frac{1}{\lambda+a_1}\rho_1\beta_{ft}(r_{f0}-f^{d*}) - \frac{1}{\lambda+a_2}\rho_1\beta_{zt}(r_{z0}-z^{d*}) - \frac{1}{2\lambda}(1-\phi^{d*})k_1f^{d*2}$$

$$J_m^{d*} = \frac{1}{\lambda}\rho_2(a_t + \theta_{ft}f^{d*} + \theta_{zt}z^{d*}) - \frac{1}{\lambda+a_1}\rho_2\beta_{ft}(r_{f0}-f^{d*}) - \frac{1}{\lambda+a_2}\rho_2\beta_{zt}(r_{z0}-z^{d*}) - \frac{1}{2\lambda}k_2z^{d*2}$$

$$- \frac{1}{2\lambda}\phi^{d*}k_1f^{d*2} + \frac{1}{\lambda}\rho_3(a_e + \theta_{ze}z^{d*}) - \frac{1}{\lambda+a_2}\rho_3\beta_{ze}(r_{z0}-z^{d*}) - \frac{1}{\lambda+a_1}\rho_3\beta_{fe}(r_{f0}-f^{d*})$$

$$- \frac{1}{2\lambda}\phi^{d*}k_1f^{d*2} + \frac{1}{\lambda}\rho_3(a_e + \theta_{ze}z^{d*}) - \frac{1}{\lambda+a_2}\rho_3\beta_{ze}(r_{z0}-z^{d*}) - \frac{1}{\lambda+a_1}\rho_3\beta_{fe}(r_{f0}-f^{d*})$$

(18)

**Property 1.** Under the condition that all members accept the cost-sharing contract, the cost-sharing rate is positively related to the marginal profit of the traditional channel and online channel respectively ($d\phi^{d*}/d\rho_2 > 0$, $d\phi^{d*}/d\rho_3 > 0$). This implies that the higher the manufacturer's marginal profit is, the more service quality cost the manufacturer will bear, increasing the motivation for the retailer's service quality improvement and obtaining more sales revenue. Although the increase of the manufacturer's marginal profit means that retailer will invest more in the service quality improvement, the manufacturer bears more service costs (Proposition 2), but after trading off between the sales revenue and the costs, the manufacturer's profit will still increase. On the contrary, the cost-sharing rate is negatively correlated with the retailer's profit margin ($d\phi^{d*}/d\rho_1 < 0$). This implies that the higher the retailer's marginal profit, the higher the retailer's service quality improvement cost (Proposition 2).

**Property 2.** In decentralized decision making, the higher the marginal profit $\rho_2$ and $\rho_3$, the higher the product quality level. The higher the marginal profit $\rho_1$, the higher the service quality level.

Property 2 shows that the marginal profit has a great impact on members' quality investment decisions. If the marginal profit is larger, the enterprise will be more motivated to improve the quality. The greater the marginal profit of the manufacturer, the more active it will be to support the retailer to improve the service quality level. On the contrary, if the marginal profit is smaller, the lower the support for service quality improvement will be.

*3.3. Model C*

Under centralized decision making, the manufacturer and the retailer are regarded as two departments in an enterprise which maximizes the profit of the enterprise by designing the optimal service quality $f(t)$ and product quality $z(t)$. The superscript $c$ indicates centralized decision making. The present value of supply chain profit under centralized decision making is

$$J_{mr}^c = \int_0^\infty e^{-\lambda t}[(\rho_2+\rho_1)d_t(t) + \rho_3 d_e(t) - \frac{1}{2}k_1f^2(t) - \frac{1}{2}k_2z^2(t)]dt$$

(19)

**Proposition 4.** *Under centralized decision making, the manufacturer's optimal product quality level and the retailer's optimal service quality level, respectively, are as follows:*

$$f^{c*} = \frac{(\rho_2+\rho_1)(\theta_{ft}+\beta_{ft})(\lambda+\alpha_1) + \rho_3\beta_{fe}\lambda - (\rho_2+\rho_1)\beta_{ft}\alpha_1}{(\lambda+\alpha_1)k_1}$$

$$z^{c*} = \frac{(\rho_2+\rho_1)(\theta_{zt}+\beta_{zt})(\lambda+\alpha_2) + \rho_3(\theta_{ze}+\beta_{ze})(\lambda+\alpha_2) - (\rho_2+\rho_1)\beta_{zt}\alpha_2 - \rho_3\beta_{ze}\alpha_2}{(\lambda+\alpha_2)k_2}$$

(20)

The reference service quality path is given by

$$r_f^{c*} = (r_{f0} - f^{c*})e^{-a_1 t} + f^{c*} \tag{21}$$

The reference product quality path is given by

$$r_z^{c*} = (r_{z0} - z^{c*})e^{-a_2 t} + z^{c*} \tag{22}$$

The proof of Proposition 4 is similar to the proof of Proposition 1 and is omitted.

Put the above conclusions into Equation (19); the profit function of the supply chain can be simplified as follows:

$$
\begin{aligned}
J_{mr}^{c*} = {} & \frac{1}{\lambda}(\rho_2 + \rho_1)(a_t + \theta_{ft} f^{c*} + \theta_{zt} z^{c*}) - \frac{1}{\lambda + a_1}(\rho_2 + \rho_1)\beta_{ft}(r_{f0} - f^{c*}) - \frac{1}{\lambda + a_1}\rho_3 \beta_{fe}(r_{f0} - f^{c*}) \\
& - \frac{1}{\lambda + a_2}(\rho_2 + \rho_1)\beta_{zt}(r_{z0} - z^{c*}) + \frac{1}{\lambda}\rho_3(a_e + \theta_{ze} z^{c*}) - \frac{1}{\lambda + a_2}\rho_3 \beta_{ze}(r_{z0} - z^{c*}) - \frac{1}{2\lambda}k_1 f^{c*2} - \frac{1}{2\lambda}k_2 z^{c*2}
\end{aligned}
\tag{23}
$$

## 4. Comparative Analysis

In order to facilitate analysis and comparison, the optimal values in three decision making scenarios are listed in Table 2.

**Table 2.** Optimal decisions under different decision scenarios.

| | Model D | Model DC | Model C |
|---|---|---|---|
| $f$ | $f^{n*} = \dfrac{\rho_1(\theta_{ft} + \beta_{ft})(\lambda + \alpha_1) - \rho_1 \beta_{ft} \alpha_1}{(\lambda + \alpha_1)k_1}$ | $f^{d*} = \dfrac{\rho_1(\theta_{ft} + \beta_{ft})(\lambda + \alpha_1) - \rho_1 \beta_{ft} \alpha_1}{(1 - \phi^d)(\lambda + \alpha_1)k_1}$ | $f^{c*} = \dfrac{(\rho_2 + \rho_1)(\theta_{ft} + \beta_{ft})(\lambda + \alpha_1) + \rho_3 \beta_{fe}\lambda - (\rho_2 + \rho_1)\beta_{ft}\alpha_1}{(\lambda + \alpha_1)k_1}$ |
| $z$ | $z^{n*} = \dfrac{\left\{ \begin{array}{l} \rho_2[(\alpha_2 + \lambda)\theta_{zt} + \lambda\beta_{zt}] \\ + \rho_3[(\alpha_2 + \lambda)\theta_{ze} + \lambda\beta_{ze}] \end{array} \right\}}{(\lambda + \alpha_2)k_2}$ | $z^{d*} = \dfrac{\left\{ \begin{array}{l} \rho_2[(\alpha_2 + \lambda)\theta_{zt} + \lambda\beta_{zt}] \\ + \rho_3[(\alpha_2 + \lambda)\theta_{ze} + \lambda\beta_{ze}] \end{array} \right\}}{(\lambda + \alpha_2)k_2}$ | $z^{c*} = \dfrac{[(\rho_2 + \rho_1)(\theta_{zt} + \beta_{zt})(\lambda + \alpha_2) + \rho_3(\theta_{ze} + \beta_{ze})(\lambda + \alpha_2) - (\rho_2 + \rho_1)\beta_{zt}\alpha_2 - \rho_3\beta_{ze}\alpha_2]}{(\lambda + \alpha_2)k_2}$ |

After analysis and comparison, we can obtain the following conclusions:

**Corollary 1.** *The optimal service quality of the retailer under three scenarios follows the following relationship:*

$$
\begin{gathered}
\text{(i) } f^{c*} > f^{d*} \\
\text{(ii) if } \ 2B > A \text{, then } \ f^{d*} > f^{n*} \\
\text{(iii) if } \ 2B \le A \text{, then } \ f^{d*} = f^{n*}
\end{gathered}
\tag{24}
$$

Proof (See Appendix C.1).

**Corollary2.** *The optimal product quality of the manufacturer under three scenarios follows the following relationship:*

$$z^{c*} > z^{d*} = z^{n*} \tag{25}$$

Proof (See Appendix C.2).

By comparing the optimal decisions under three scenarios, it can be found that the optimal decisions of product quality and service quality under the centralized decision are higher than those under the decentralized decision making scenario, respectively. When the condition $2B > A$ is satisfied, the optimal decision of service quality with a cost-sharing contract is greater than the corresponding value without a cost-sharing contract, while for the product quality, there is no improvement under the two decision scenarios.

The optimal profit of the supply chain under the three scenarios is compared and analyzed, and the following conclusions are obtained.

**Corollary 3.** *The profit of the supply chain under the three scenarios follows the following relationship:*

$$(i)\ J_{mr}^{\ c*} > J_{mr}^{\ d*}$$

$$(ii)\ \text{if}\ \ 2B > A\ ,\ \text{then}\ \ J_{mr}^{\ d*} > J_{mr}^{\ n*} \qquad (26)$$

$$(iii)\ \text{if}\ \ 2B \leq A\ ,\ \text{then}\ \ J_{mr}^{\ d*} = J_{mr}^{\ n*}$$

Proof (See Appendix C.3).

## 5. Supply Chain Coordination Mechanism (Model CS)

From the above comparative analysis, it can be concluded that optimal decisions and profit of the supply chain under centralized decision making are greater than that under decentralized decision making. It is necessary to design an effective coordination mechanism to distribute the supply chain overall profit of centralized decision making, so that the relationship between the manufacturer and the retailer can attribute to achieve Pareto optimality. To achieve the above objective, the supply chain is then coordinated by introducing a cost-sharing coordination mechanism. In the cost-sharing coordination mechanism, both the manufacturer and the retailer adopt the optimal decisions of centralized decision making. At the same time, the leading manufacturer distributes the total profit of the supply chain in the centralized decision making by making a reasonable service cost-sharing rate $\phi^{cs}$. According to above analysis, the present value profits of the manufacturer and the retailer can be expressed as follows:

$$J_m^{cs*} = \frac{1}{\lambda}\rho_2(a_t + \theta_{ft}f^{c*} + \theta_{zt}z^{c*}) - \frac{1}{\lambda + a_1}\rho_2\beta_{ft}(r_{f0} - f^{c*}) - \frac{1}{\lambda + a_1}\rho_3\beta_{fe}(r_{f0} - f^{c*}) - \frac{1}{2\lambda}k_2 z^{c*2}$$

$$- \frac{1}{\lambda + a_2}\rho_2\beta_{zt}(r_{z0} - z^{c*}) - \frac{1}{2\lambda}\phi^{cs*}k_1 f^{c*2} + \frac{1}{\lambda}\rho_3(a_e + \theta_{ze}z^{c*}) - \frac{1}{\lambda + a_2}\rho_3\beta_{ze}(r_{z0} - z^{c*}) \qquad (27)$$

$$J_r^{cs*} = \frac{1}{\lambda}\rho_1(a_t + \theta_{ft}f^{c*} + \theta_{zt}z^{c*}) - \frac{1}{\lambda + a_1}\rho_1\beta_{ft}(r_{f0} - f^{c*}) - \frac{1}{\lambda + a_2}\rho_1\beta_{zt}(r_{z0} - z^{c*}) - \frac{1}{2\lambda}(1 - \phi^{cs*})k_1 f^{c*2} \qquad (28)$$

By adding Equations (27) and (28) together, we obtain Equation (29), which is equal to the total profit of the supply chain of the centralized decision making. The following equation can be obtained.

$$J_{mr}^{\ cs*} = J_{mr}^{\ c*} \qquad (29)$$

Setting the service cost-sharing rate $\phi^{cs}$ will affect whether the total profit of the supply chain is reasonably distributed under the cost-sharing coordination mechanism.

$$J_{mr}^{\ c*} = \frac{1}{\lambda}(\rho_2 + \rho_1)(a_t + \theta_{ft}f^{c*} + \theta_{zt}z^{c*}) - \frac{1}{\lambda + a_1}(\rho_2 + \rho_1)\beta_{ft}(r_{f0} - f^{c*}) - \frac{1}{\lambda + a_1}\rho_3\beta_{fe}(r_{f0} - f^{c*})$$

$$- \frac{1}{\lambda + a_2}(\rho_2 + \rho_1)\beta_{zt}(r_{z0} - z^{c*}) + \frac{1}{\lambda}\rho_3(a_e + \theta_{ze}z^{c*}) - \frac{1}{\lambda + a_2}\rho_3\beta_{ze}(r_{z0} - z^{c*}) - \frac{1}{2\lambda}k_1 f^{c*2} - \frac{1}{2\lambda}k_2 z^{c*2} \qquad (30)$$

The necessary condition to achieve supply chain coordination is that the manufacturer's profit is not less than the manufacturer's optimal profit under decentralized decision making while the cost-sharing rate meets $\phi_{max}^{cs} \in [0,1]$ under the coordination mechanism.

$$J_m^{cs*} \geq J_m^{d*}, 0 \leq \phi_{max}^{cs} \leq 1 \qquad (31)$$

It is found from Equation (31) that the largest rate $\phi_{max}^{cs}$ that the manufacturer is willing to share of the service cost of the retailer is

$$\phi_{max}^{cs} = \frac{2\lambda\left\{ \begin{array}{l} [\frac{1}{\lambda}\rho_2\theta_{ft} + \frac{1}{\lambda + a_1}\rho_2\beta_{ft} + \frac{\rho_3\beta_{fe}}{(\lambda + a_1)(\lambda + \mu)}](f^{c*} - f^{d*}) + [\frac{1}{\lambda}\rho_2\theta_{zt} + \frac{1}{\lambda + a_2}\rho_2\beta_{zt} \\ + \frac{1}{\lambda}\rho_3\theta_{ze} + \frac{1}{\lambda + a_2}\rho_3\beta_{ze}](z^{c*} - z^{d*}) + \frac{1}{2\lambda}\phi^{d*}k_1 f^{d*2} + \frac{1}{2\lambda}k_2(z^{d*2} - z^{c*2}) \end{array} \right\}}{k_1 f^{c*2}} \qquad (32)$$

In the same way, the necessary condition for the retailer to accept the coordination mechanism is that the service cost-sharing rate satisfies $\phi_{min}^{cs} \in [0,1]$, and the retailer's profit is not less than the optimal profit of the retailer under the decentralized decision.

$$J_r^{cs*} \geq J_r^{d*}, 0 \leq \phi_{min}^{cs} \leq 1 \tag{33}$$

It is found from Equation (33) that the service cost-sharing rate that the retailer requires the manufacturer to share cannot be lower than $\phi_{min}^{cs}$:

$$\phi_{min}^{cs} = 1 - \frac{2\lambda \left\{ \begin{array}{l} [\frac{1}{\lambda}\rho_1\theta_{ft} + \frac{1}{\lambda + a_1}\rho_1\beta_{ft}](f^{c*} - f^{d*}) + [\frac{1}{\lambda}\rho_1\theta_{zt} \\ + \frac{1}{\lambda + a_2}\rho_1\beta_{zt}](z^{c*} - z^{d*}) + \frac{1}{2\lambda}(1 - \phi^{d*})k_1 f^{d*2} \end{array} \right\}}{k_1 f^{c*2}} \tag{34}$$

Moreover, it is easy to get $\phi_{max}^{cs} - \phi_{min}^{cs} > 0$; thus, we obtain the following proposition.

**Proposition 5**. *If the service cost-sharing rate satisfies $\phi^{cs} \in [\phi_{max}^{cs}, \phi_{min}^{cs}]$, $\phi_{min}^{cs} \in [0,1]$ and $\phi_{max}^{cs} \in [0,1]$, it can realize the coordination of the supply chain system.*

Proposition 5 shows that if the service cost-sharing rate satisfies $\phi^{cs} \in [\phi_{max}^{cs}, \phi_{min}^{cs}]$, $\phi_{min}^{cs} \in [0,1]$ and $\phi_{max}^{cs} \in [0,1]$, both the manufacturer and the retailer adopt the optimal strategies of the centralized decision. The value of $\phi^{cs}$ depends on their bargaining power in the supply chain.

## 6. Numerical Analysis

Due to the complexity of the model, some analytical properties are difficult to analyze. This section uses numerical examples to compare the performance of firms under different decision making scenarios, and further analyzes the impact of important parameters on firms' decision making and profits in order to obtain corresponding management insights.

We consider the following values: $\lambda = 0.1$, $\rho_1 = 0.5$, $\rho_2 = 1$ and $\rho_3 = 1$, $\alpha_1 = 1$, $\alpha_2 = 2$, $\beta_{ft} = 0.1$, $\beta_{zt} = 0.2$, $\beta_{ze} = 0.1$, $\beta_{fe} = 0.05$, $\theta_{ft} = 0.5$, $\theta_{zt} = 0.8$, $\theta_{ze} = 0.8$, $k_1 = 1$, $k_2 = 1$, $r_{z0} = 3$, $r_{f0} = 2$, $a_t = 2$, $a_e = 2$. To obtain qualitative insights into how the present value of profit varies as the service cost-sharing rate $\phi^{cs}$ varies under the coordination mechanism scenario and decentralized decision scenario, we keep the other parameters fixed. The relationships are shown in Figure 2.

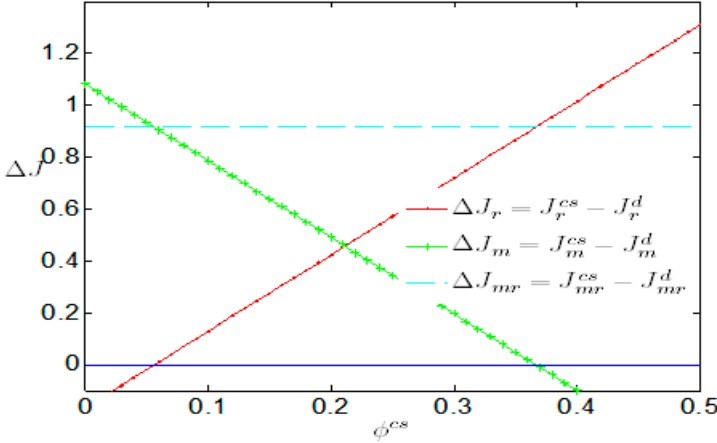

**Figure 2.** The impact of service cost-sharing rate on profits.

As shown in Figure 2, the manufacturer's profit difference under the two different scenarios decreases with the increase of the service cost-sharing rate $\phi^{cs}$. In the range of (0,0.37), the profit of the manufacturer is not less than that of the manufacturer in decentralized decision making. While in the range of (0.37,1), the profit of the manufacturer is less than that of the manufacturer in decentralized decision making. The profit difference of the retailer under the two different scenarios increases with the increase of service cost-sharing rate $\phi^{cs}$. In the range of (0.05,1), the retailer's profit is not less than the retailer's profit under decentralized decision. In the range of (0,0.05), the retailer's profit is smaller than the retailer's profit under decentralized decision making. Therefore, it is found that when the service cost-sharing rate is in the range of (0.05,0.37), the profits of the members under the coordination mechanism scenario are not less than those of the members under the decentralized decision making scenario.

Next, we present the value of members' profits as the effectiveness parameters vary under the different scenarios. The relationships are shown in Figure 3.

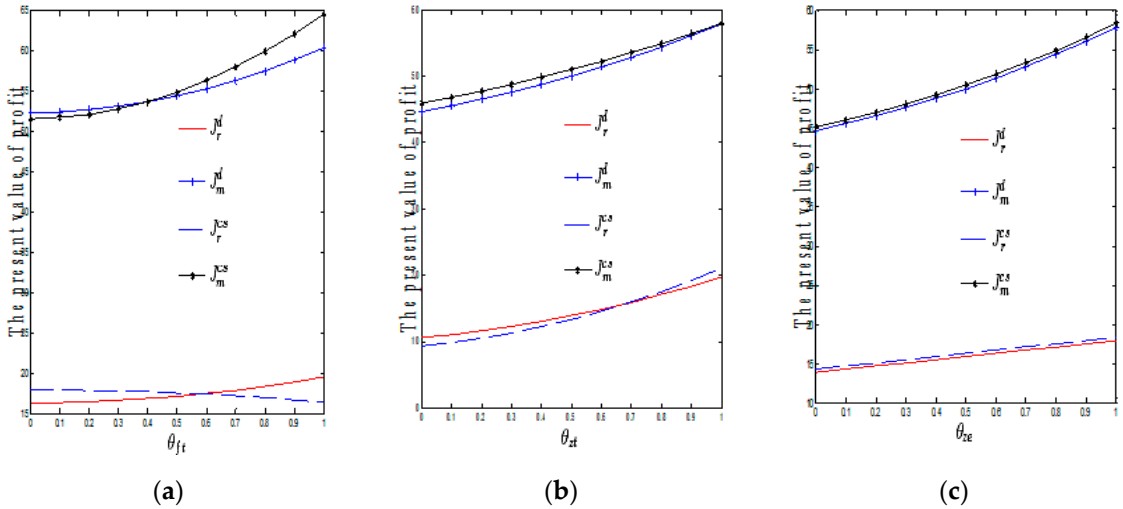

|     |     |     |
| :-: | :-: | :-: |
| (**a**) | (**b**) | (**c**) |

**Figure 3.** The impact of effectiveness parameters on profits. (**a**)The impact of $\theta_{ft}$ on profits (**b**) The impact of $\theta_{zt}$ on profits (**c**) The impact of $\theta_{ze}$ on profits.

As shown in Figure 3a, the members' profits increase with the increase of the effectiveness parameter $\theta_{ft}$. This is because the more consumers pay attention to service quality, the more the retailer invests in service quality, the more the market demand of products can be stimulated, and thus the greater the retailer's profit. Although the manufacture shares a certain rate of the retailer's service cost, the improvement of service quality increases the demand of both traditional and online channels. Because the sales revenue generated is greater than the cost borne by the manufacturer, the manufacturer's profit increases. It can also be seen from Figure 3a that when $\theta_{ft}$ is small, the retailer's profit under the cost-sharing coordination mechanism ( $\phi_{cs} = 0.2$ ) is larger than that under the decentralized decision making. With the increase of $\theta_{ft}$, when it exceeds a certain range, the retailer's profit cannot be improved under the service cost-sharing rate $\phi_{cs} = 0.2$. Meanwhile, for the manufacturer, when $\theta_{ft}$ is small, the manufacturer's profit cannot be improved under the service cost-sharing rate $\phi_{cs} = 0.2$. With the increase of $\theta_{ft}$, when it exceeds a certain range, the profit of the manufacturer is greater than that of decentralized decision making.

As shown in Figure 3b, the members' profits increase with the increase of the effectiveness parameters $\theta_{zt}$. This is because the more consumers pay attention to product quality, the more the manufacturer invests in product quality, the more market demand of products can be stimulated, and thus the greater the profits of the manufacturer and retailer. It can also be seen from Figure 3b that for the retailer, when the effectiveness parameters $\theta_{zt}$ is small, the retailer's profit cannot be

improved under the service cost-sharing rate $\phi^{cs} = 0.2$. Meanwhile, for the manufacturer, when the effectiveness parameters $\theta_{zt}$ exceeds a certain range, the manufacturer's profit cannot be improved under the service cost-sharing rate $\phi^{cs} = 0.2$.

As shown in Figure 3c, the members' profits increase with the increase of the effectiveness parameters $\theta_{ze}$, and the members' profits can be improved under the service cost-sharing rate $\phi^{cs} = 0.2$ no matter how $\theta_{ze}$ changes.

In Figure 4, we present the value of members' profits along with the initial reference quality varies under the different scenarios.

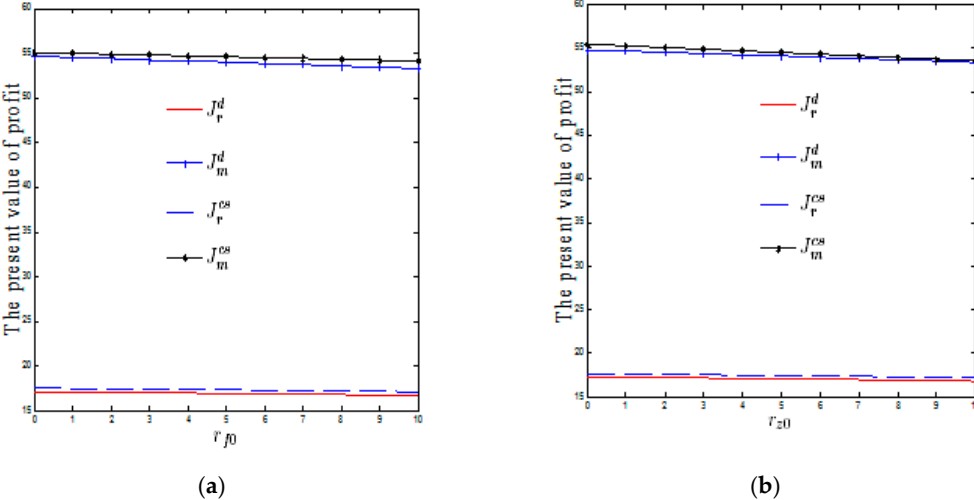

(**a**)                                             (**b**)

**Figure 4.** The impact of initial reference quality on profits. (**a**)The impact of $r_{f0}$ on profits. (**b**)The impact of $r_{z0}$ on profits.

By using Figure 4, we can obtain the following result. The profits of the members decrease with the increase of initial reference quality under the two different scenarios. This is because when the initial reference quality is low, the quality level provided by the manufacturer or the retailer can easily meet the consumers' requirements, which can have a positive impact on demand and improve the profits of the members. When the initial reference quality is high, the quality level has difficulty meeting the requirements of consumers, which has a negative impact on demand and reduces the profits of the members.

Figure 5a shows the impact of the service quality reference effect on profits. When the initial reference service quality is low, the manufacturer's profit increases with the increase of the service quality reference effect, while the initial reference service quality is high, the manufacturer's profit decreases with the increase of service quality reference effect. No matter what the initial reference service quality is, the service quality reference effect is not conducive to the retailer. Only when the initial reference service quality is low, can the service quality reference effect benefit the manufacturer. At this time, as the leader of the supply chain, the manufacturer should share more service cost and encourage the retailer to improve the service quality.

Figure 5b shows the impact of the service quality spillover effect on profits. When the initial reference service quality is low, the manufacturer's profit increases with the increase of service quality spillover effect, while the retailer's profit decreases with the increase of the service quality spillover effect. It can be concluded that the spillover effect of the service quality is not conducive to the retailer and the manufacturer. At this time, as the leader of the supply chain, the manufacturer should share more service cost to make up for the retailer's loss caused by the spillover effect, so as to encourage the retailer to improve service quality.

Figure 5c shows the impact of the online product quality reference effect on profits. When the initial reference product quality is low, the profit of the manufacturer increases with the increase of the online product quality reference effect, while when the initial reference product quality is high,

the profit of the manufacturer decreases with the increase of the online product quality reference effect. It can be inferred that when the initial reference product quality is low, the online product quality reference effect is conducive to the retailer and the manufacturer, at this time, the manufacturer should increase investment in product quality. When the initial reference quality product is high, the retailer should bear more service cost to reduce the burden of the manufacturer.

Figure 5d shows the impact of offline product quality reference effect on profits. When the initial reference product quality is low, the profit of manufacturer and retailer increases with the increase of offline product quality reference effect, while when the initial reference product quality is high, the profit of the manufacturer and retailer decrease with the increase of offline product quality reference effect. It can be inferred that when the initial reference product quality is low, the offline product quality reference effect is conducive to the retailer and the manufacturer, at this time, the manufacturer should increase investment in product quality.

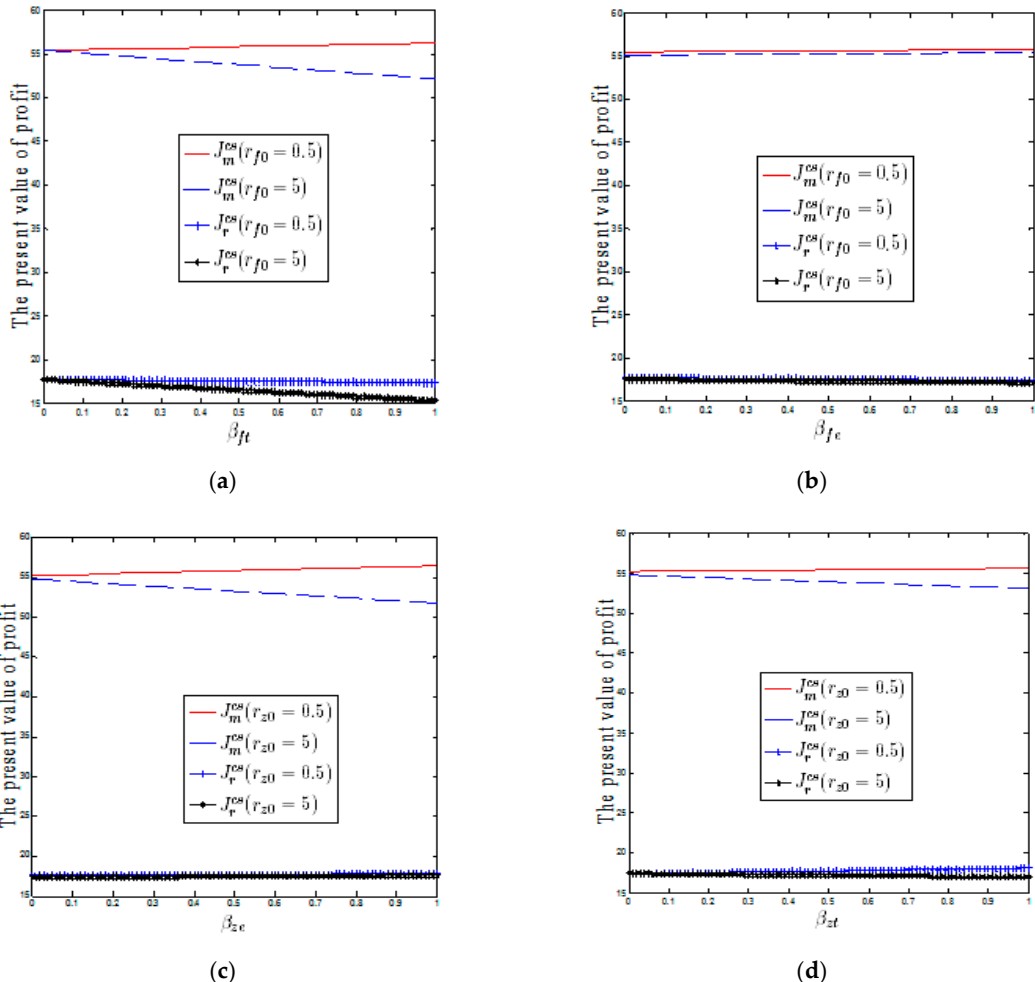

**Figure 5.** Impact of reference effect on profits. (**a**) The impact of $\beta_{ft}$ on profits. (**b**) The impact of $\beta_{fe}$ on profits. (**c**) The impact of $\beta_{ze}$ on profits. (**d**) The impact of $\beta_{zt}$ on profits.

In Figure 6, we present the change of reference quality with time along with the initial reference quality varies under the different scenarios.

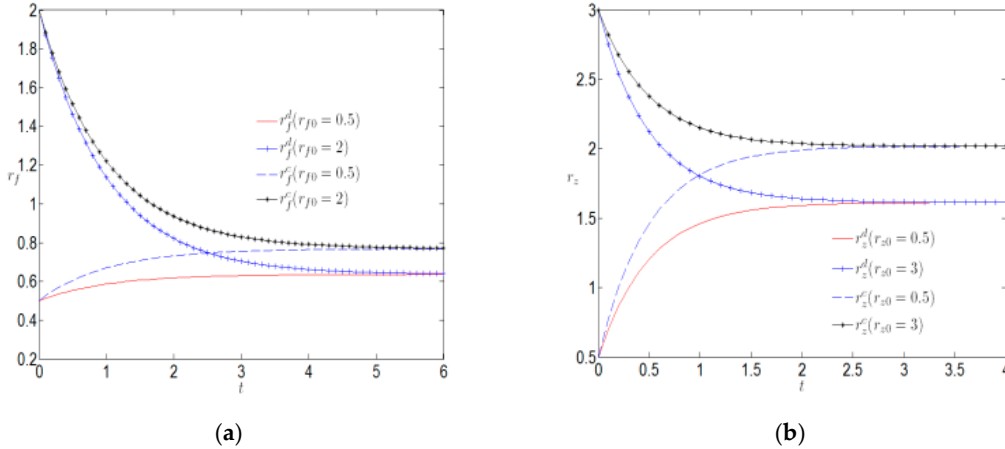

**Figure 6.** The change of reference quality with time. (**a**) The change of $r_f$ with time. (**b**)The change of $r_z$ with time.

It is found from Figure 6 that, under the two different scenarios, when the initial reference quality is lower ($r_{z0} = 2$ or $r_{z0} = 0.5$), the reference quality will increase with time. When the initial reference quality is higher ($r_{f0} = 2$ or $r_{z0} = 3$), the reference quality will decrease with time. The reference quality with time will tend to the same stable value, which is the optimal quality strategy. This implies that a different initial reference quality will affect the paths of the change of the reference quality changing with time, but will not affect the final stable value. In addition, it is found that the stability value of the reference quality under the coordination mechanism is greater than that under the decentralized decision making.

## 7. Conclusions and Discussion

We consider a dual channel supply chain system consisting of a manufacturer and a retailer in which the manufacturer sells a final product through both a traditional channel retailer and their own online channel in parallel. Based on the quality reference effect and the service quality spillover effect, the paper builds decision models under the different scenarios, compares the optimal decisions and profits, and designs the cost-sharing coordination mechanism. The following conclusions are obtained.

In the three decision making scenarios, the optimal value of product quality and service quality is the highest under a centralized decision making. In the case of decentralized decision making, if the marginal profit satisfies a certain relationship, the cost sharing contract can encourage the retailer to improve service quality, but it cannot promote the manufacturer to improve product quality. In the case of centralized decision, the overall profit of supply chain is greater than that in a decentralized decision. If the marginal profit satisfies a certain relationship, the manufacturer's profit, the retailer's profit and the supply chain's overall profit under the decentralized cost sharing decision making are all greater than these in a no cost sharing contract. Given the above analysis, the vertical integration of supply chain can reduce the double marginal effect and improve the profits of manufacturer and retailer. At the same time, under the competitive market environment to date, the vertical integration can improve the product quality and service quality and thus improve the competitiveness of enterprises. There is a saying: "today's market competition is not the competition between enterprises, but the competition between supply chains.". So vertical alliance is an important way of sustainable development for enterprises. Otherwise, they will be eliminated by the market. Dong Mingzhu, chairman of Gree Electric, emphasized that the cooperation between Gree (an excellent manufacturer) and Suning (an excellent seller) is a strong combination of technology and service. Gree is committed to mastering the core technology, using its own R & D technology to make the world fall in love with made in China, while Suning constantly improves its service [42].

Under the decentralized decision making, the cost-sharing rate provided by the manufacturer is positively related to the marginal profit of the manufacturer's traditional and online channels. The higher the manufacturer's marginal profit is, the more manufacturer will bear the service cost, meaning that the retailer will further increase the enthusiasm of service quality investment, the manufacturer's income increase. On the contrary, the cost-sharing rate is negatively related to the retailer's marginal profit. The higher the retailer's marginal profit is, the more active the retailer is in improving the service quality. Meanwhile, according to the contract, the manufacturer will bear more service cost, which to a certain extent affects the enthusiasm of the manufacturer to share the retailer's service improvement cost. Therefore, the manufacturer adopts advanced technology, optimizes the production process, and realizes the purpose of reducing the production cost. It is possible for the manufacturer to outsource the logistics business to a professional third-party logistics enterprise, reduce the cost of the circulation link, so as to improve the marginal profit of the manufacturer's unit products. Second, manufacturer design products to meet the needs of consumers' personality, and then improve the value of products. Otherwise, manufacturers will lose the chance to make more profits.

When the initial reference service quality is low, the consumer service quality reference effect is beneficial to the manufacturer. At this time, the manufacturer should share more service cost, so as to encourage the retailer improvement of service quality. The spillover effect of service quality is not conducive to the retailer and the manufacturer. Therefore, it is necessary for the manufacturer to share more service cost to make up for the loss of the retailer caused by spillover effect, so as to enhance the enthusiasm of the retailer to improve service quality. Otherwise, retailers may lower quality of service, which may affect the sustainability of the cooperation. When the initial reference product quality is low, both online and offline product quality reference effects are beneficial to the retailer and the manufacturer. Then, the manufacturer may increase investment in product quality. When the initial reference product quality is high, it is necessary for the retailer to bear more service cost to reduce the manufacturer economic burden caused by the adverse product quality reference effect on the manufacturer. Otherwise, it may affect the manufacturer's enthusiasm for product quality improvement, which will lead to the breakdown of cooperation and the loss of retailers' profits.

If the rate of cost-sharing satisfies a certain relationship, and the manufacturer and retailer adopt the optimal quality decisions of the centralized decision respectively, the supply chain system reaches the coordination state, which can realize the Pareto improvement for the manufacturer and retailer. On the contrary, if the cost-sharing ratio provided by the manufacturer is not within the reasonable range that the retailer can accept, the retailer will refuse to cooperate with the manufacturer, that is, the retailer will not take the optimal strategy of centralized decision making, and the manufacturer's profit will not be improved. The coordination mechanism proposed in this paper is an effective way to realize the supply chain alliance, which can have a positive guiding significance for the business decision making. At the same time, maintaining a long-term stable cooperative relationship can not only improve the present value of their profits, but also ensure the stability of the supply chain system.

Different initial reference quality will only affect the track of reference quality changing with time, and reference quality will tend to be stable with time. The stable (or final) reference quality will not be affected by the initial reference quality. And the stable reference quality is equal to the optimal quality. It can be concluded that before the supply chain reaches a stable state, the reference effect can have an impact on the profits of the supply chain. After reaching a stable state, the reference effect has no impact on the profits of the supply chain.

The quality characteristics of products generally have six aspects: performance, life (durability), reliability and maintainability, safety, adaptability and economy. There is no doubt that consumers attach great importance to product quality. Therefore, manufacturers should actively improve product quality by introducing advanced technology or R & D investment, and then expand market share, so as to obtain greater present value of profits. The improvement of product quality is conducive to the formation of good goodwill, which is the basis for the sustainable development of

enterprises. In addition, at present consumers prefer green products and the government issues relevant environmental protection policies, so in the process of production, enterprises should not only meet the standards of product quality characteristics, but also meet the ecological standards of products, which is conducive to the sustainable development of enterprises. Otherwise, it will cause unnecessary economic loss. According to the Ministry of ecology and environment of China in 2019, due to environmental protection problems, Mercedes Benz (China) Automobile Sales Co., Ltd. recalled 302 imported Benz vehicles, which were mixed with exhaust catalytic converters, resulting in the failure to meet the requirements of vehicle emission standards, etc., which may lead to the alarm of on-board diagnosis system [43]. In the context of circular economy, the government not only promulgates environmental laws and regulations, but also should strengthen the publicity of circular economy, advocate ecological culture, so as to improve the public's awareness of environmental protection and promote ecological consumption.

There are still some limitations and deficiencies in this paper, which need to be studied and expanded in the future. First of all, this paper assumes homogeneous consumers, which have the behavioral characteristics of quality reference effect. In real life, some consumers are not sensitive to quality reference effect. In the future, we can study the impact of consumers heterogeneity on supply chain quality strategy. Secondly, the research of this paper is based on the dual channel supply chain, and more complex channel structure can be considered in the future. For example, in the same sales area the manufacturer may sell products through multiple retailers at the same time. Thirdly, considering the influence of competition behavior between decision-makers on supply chain quality strategy and supply chain coordination is the future direction worthy of further study. In addition, future research can also focus on the behavior factors of decision makers.

**Author Contributions:** Conceptualization, Z.X. and X.B.; methodology, Z.X. and X.F.; software, Z.X.; formal analysis, X.B.; investigation, X.F.; writing—original draft, X.F.; writing—review and editing, Z.X. and L.Y.; project administration X.B. All authors have read and agreed to the published version of the manuscript.

**Funding:** This research was funded by the Higher Education Humanities and Social Science Research Project of Jiangxi Province(No.GL18110),the Postdoctoral Research Project of Jiangxi Province(No.2019KY39),the National Natural Science Foundation of China (No.71561018),the Program of Natural Science Foundation of Jiangxi Province (No.20171BAA218002), and the Science Technology Program of Education Department of Jiangxi Province (No.GJJ171003).

**Conflicts of Interest:** The authors declare no conflflict of interest.

## Appendix A

**Proof of Proposition 1.** First, taking Equations (1) into account, the optimal decision problem of the retailer is expressed as $\max_{f>0} J_r^n$. Using the maximum principle, its Hamilton function is

$$H_r^n = \rho_1[a_t + \theta_{ft}f + \beta_{ft}(f - r_f) + \theta_{zt}z + \beta_{zt}(z - r_z)] - \frac{1}{2}k_1 f^2 + X_{rf}\alpha_1(f - r_f) \tag{A1}$$

The retailer's optimal decision problem needs to satisfy the following conditions

$$\frac{dH_r^n}{df} = \rho_1(\theta_{ft} + \beta_{ft}) - k_1 f + X_{rf}\alpha_1 = 0 \tag{A2}$$

$$\dot{X}_{rf}(t) = \lambda X_{rf} - \frac{dH_r^n}{dr_f} = (\lambda + \alpha_1)X_{rf} + \rho_1\beta_{ft} \tag{A3}$$

From Equation (A2), we obtain

$$f^n = \frac{\rho_1(\theta_{ft} + \beta_{ft}) + X_{rf}\alpha_1}{k_1} \tag{A4}$$

By solving the differential Equation (A3), we obtain

$$X_{rf}(t) = c_1 e^{(\lambda + \alpha_1)t} - \frac{\rho_1 \beta_{ft}}{\lambda + \alpha_1} \tag{A5}$$

Substituting Equation (A4) into Equation (A3), we obtain

$$f^n = \frac{\rho_1(\theta_{ft} + \beta_{ft})(\lambda + \alpha_1) + (\lambda + \alpha_1)\alpha_1 c_1 e^{(\lambda + \alpha_1)t} - \alpha_1 \rho_1 \beta_{ft}}{k_1(\lambda + \alpha_1)} \tag{A6}$$

Because when $t \to \infty$, the service quality of the retailer is limited, so

$$\lim_{t \to \infty} f^n(t) < \infty \tag{A7}$$

From the above Equation, it can be judged that $c_1 = 0$ in Equation (A6). After rearrangement, the optimal service quality of the retailer is

$$f^{n*} = \frac{\rho_1(\theta_{ft} + \beta_{ft})(\lambda + \alpha_1) - \alpha_1 \rho_1 \beta_{ft}}{k_1(\lambda + \alpha_1)} \tag{A8}$$

Substituting Equation (A8) into Equation (1), the reference service quality path is obtained as follows

$$r_f^{\,n}(t) = (r_{f0} - f^{n*})e^{-a_1 t} + f^{n*} \tag{A9}$$

Taking Equation (2) into account, the optimal decision problem of the manufacturer is expressed as $\max_{z>0} J_m$.

Using the maximum principle to construct the Hamilton function

$$H_m^{\,n} = \rho_2[a_t + \theta_{ft}f + \beta_{ft}(f - r_f) + \theta_{zt}z + \beta_{zt}(z - r_z)] + \rho_3[a_e + \theta_z z + \beta_{ze}(z - r_z) + \beta_{fe}(f - r_f)]$$
$$- \frac{1}{2}k_2 z^2 + X_{mz}\alpha_2(z - r_z) \tag{A10}$$

Similarly, the manufacturer's optimal decision problem needs to satisfy the following conditions

$$\frac{dH_m^{\,n}}{dz} = \rho_2(\theta_{zt} + \beta_{zt}) + \rho_3(\theta_{ze} + \beta_{ze}) - k_2 z + X_{mz}\alpha_2 = 0 \tag{A11}$$

$$\dot{X}_{mz}(t) = \lambda X_{mz} - \frac{dH_m^{\,n}}{dr_z} = (\lambda + \alpha_2)X_{mz} + \rho_2\beta_{zt} + \rho_3\beta_{ze} \tag{A12}$$

From Equation (A11), we obtain

$$z^n = \frac{\rho_2(\theta_{zt} + \beta_{zt}) + \rho_3(\theta_{ze} + \beta_{ze}) + X_{mz}\alpha_2}{k_2} \tag{A13}$$

By solving the differential Equation (A12), we obtain

$$X_{mz}(t) = c_2 e^{(\lambda + \alpha_2)t} - \frac{\rho_2\beta_{zt} + \rho_3\beta_{ze}}{\lambda + \alpha_2} \tag{A14}$$

Substituting Equation (A14) into Equation (A13), we obtain

$$z^n = \frac{\left\{\begin{array}{l}(\lambda + \alpha_2)\alpha_2 c_3 e^{(\lambda + \alpha_2)t} + (\lambda + \alpha_2)[\rho_2(\theta_{zt} + \beta_{zt}) \\ + \rho_3(\theta_{ze} + \beta_{ze})] - (\rho_2\beta_{zt} + \rho_3\beta_{ze})\alpha_2\end{array}\right\}}{k_2(\lambda + \alpha_2)\lambda} \tag{A15}$$

Because when $t \to \infty$, the quality of products produced by the manufacturer is limited, so

$$\lim_{t \to \infty} z^n(t) < \infty \tag{A16}$$

From the above Equation, it can be judged that $c_2 = 0$ in Equation (A15). After rearrangement, the optimal product quality designed by the manufacturer is

$$z^{n*} = \frac{(\alpha_2 + \lambda)[\rho_2(\theta_{zt} + \beta_{zt}) + \rho_3(\theta_{ze} + \beta_{ze})] - (\rho_2\beta_{zt} + \rho_3\beta_{ze})\alpha_2}{(\alpha_2 + \lambda)k_2} \tag{A17}$$

Similarly, the reference product quality path is obtained as follows

$$r_z^{n*}(t) = (r_{z0} - z^{n*})e^{-a_2 t} + z^{n*} \tag{A18}$$

Because of $\alpha_1 > 0$, $\alpha_2 > 0$, the reference service quality path given in Equation (A9) will converge to the steady state $f^{n*}$ when $t \to +\infty$. Similarly, the reference product quality path given in Equation (A18) will converge to the steady state $z^{n*}$ when $t \to +\infty$. □

## Appendix B

**Proof of Proposition 3.** Put the conclusion of Proposition 2 into Equation (13) (manufacturer's profit $J_m^d$), and make $\dfrac{dJ_m^d}{d\phi^d} = 0$, we obtain

$$\frac{dJ_m^d}{d\phi^d} = \frac{B}{\lambda} - \frac{A}{2\lambda} - \frac{A}{\lambda}\frac{\phi^d}{(1 - \phi^d)} \tag{A19}$$

where $B = \dfrac{\rho_2[\theta_{ft}(\lambda + \alpha_1) + \lambda\beta_{fe}] + \rho_3\lambda\beta_{fe}}{(\lambda + \alpha_1)}$, $A = \dfrac{\rho_1(\theta_{ft}\lambda + \beta_{ft}\lambda + \theta_{ft}\alpha_1)}{(\lambda + \alpha_1)}$. Due to $0 \le \phi^d \le 1$, we obtain by simplifying Equation (A19), when $2B > A$, then $\phi^d = \dfrac{2B - A}{2B + A}$, otherwise $\phi^d = 0$. □

## Appendix C

*Appendix C.1*

**Proof of Corollary 1.** *Let*

$$f^{c*} - f^{d*} = \frac{(\theta_{ft}\lambda + \beta_{ft}\lambda + \theta_{ft}\alpha_1)[(1 - \phi^d)(\rho_2 + \rho_1) - \rho_1]}{(1 - \phi^d)(\lambda + \alpha_1)k_1} + \frac{\rho_3\beta_{fe}}{(\lambda + \alpha_1)k_1} \tag{A20}$$

When $2B > A$, that is $\phi^d = \dfrac{2B - A}{2B + A}$, we obtain

$$f^{c*} - f^{d*} = \frac{A\rho_1}{2k_1} > 0 \tag{A21}$$

When $2B \le A$, that is $\phi^d = 0$, we obtain

$$f^{c*} - f^{d*} = \frac{(\theta_{ft}\lambda + \beta_{ft}\lambda + \theta_{ft}\alpha_1)\rho_2 + \rho_3\beta_{fe}}{(\lambda + \alpha_1)k_1} > 0 \tag{A22}$$

Similarly,

$$f^{d*} - f^{n*} = \frac{\rho_1(\theta_{ft} + \beta_{ft})(\lambda + \alpha_1) - \rho_1\beta_{ft}\alpha_1}{(\lambda + \alpha_1)k_1}\left(\frac{1}{1 - \phi^d} - 1\right) \tag{A23}$$

When $2B > A$, we obtain

$$f^{d*} - f^{n*} = \frac{(\theta_{ft}\lambda + \beta_{ft}\lambda + \theta_{ft}\alpha_1)\rho_2 + \rho_3\beta_{fe}}{(\lambda + \alpha_1)k_1} > 0 \tag{A24}$$

When $2B \le A$, we obtain

$$f^{d*} - f^{n*} = 0 \tag{A25}$$

□

*Appendix C.2*

**Proof of Corollary 2.** Let

$$z^{c*} - z^{d*} = \frac{\rho_1[(\lambda + \alpha_2)\theta_{zt} + \lambda\beta_{zt}]}{(\lambda + \alpha_2)k_2} > 0 \tag{A26}$$

Then

$$z^{c*} > z^{d*} \tag{A27}$$

We easily obtain

$$z^{d*} = z^{n*} \tag{A28}$$

□

*Appendix C.3*

**Proof of Corollary 3.** When $2B > A$, we obtain

$$\begin{aligned} &J_{mr}{}^{c*} - J_{mr}{}^{d*} \\ &= \frac{\rho_1}{4}(\frac{\theta_{ft}}{\lambda} + \frac{\beta_{ft}}{\lambda + \alpha_1})(f^{c*} - f^{d*}) + \rho_1(\frac{\theta_{zt}}{\lambda} + \frac{\beta_{zt}}{\lambda + a_2})(z^{c*} - z^{d*}) \end{aligned} \tag{A29}$$

When $2B \le A$, we obtain

$$\begin{aligned} &J_{mr}{}^{c*} - J_{mr}{}^{d*} \\ &= \left[\frac{\rho_2\theta_{ft}}{\lambda} + \frac{\rho_2\beta_{ft}}{\lambda + a_1} + \frac{\rho_3\beta_{fe}}{\lambda(\lambda + \alpha_1)}\right]\frac{(f^{c*} - f^{d*})}{2} + (\frac{\rho_1\theta_{zt}}{\lambda} + \frac{\rho_1\beta_{zt}}{\lambda + a_2})(z^{c*} - z^{d*}) \end{aligned} \tag{A30}$$

According to Corollary 1 and Corollary 2, we obtain $f^{c*} > f^{d*}$, $z^{c*} > z^{d*}$; then, the relationship $J_{mr}{}^{c*} > J_{mr}{}^{d*}$ can be judged.

Similarly,

When $2B > A$

$$J_{mr}{}^{d*} - J_{mr}{}^{n*} = (\frac{B}{2\lambda} + \frac{A}{4\lambda})(f^{d*} - f^{n*}) \tag{A31}$$

Because $f^{d*} > f^{n*}$, we obtain

$$J_{mr}{}^{d*} > J_{mr}{}^{n*} \tag{A32}$$

When $2B \le A$, we obtain $f^{d*} = f^{n*}$ from Corollary 1, so we obtain

$$J_{mr}{}^{d*} = J_{mr}{}^{n*} \tag{A33}$$

where $B = \lambda\left[\frac{\rho_2\theta_{ft}}{\lambda} + \frac{\rho_2\beta_{ft}}{\lambda + a_1} + \frac{\rho_3\beta_{fe}}{\lambda(\lambda + \alpha_1)}\right]$, $A = \frac{\rho_1(\theta_{ft}\lambda + \beta_{ft}\lambda + \theta_{ft}\alpha_1)}{(\lambda + \alpha_1)}$. □

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
