# Peer review of "Research on Quality Decisions and Coordination with Reference Effect in Dual-Channel Supply Chain"

_sustainability, doi:10.3390/su12062296_

Round 1

Reviewer 1 Report

I analyze the single sections:

Abstract has inappropriate structure. I suggest to answer the following aspects: - general context - novelty of the work - methodology used (describe briefly the main methods or treatments applied) - main results and related interpretations.

Introduction: This section should briefly place the study in a wide context and emphasize why it is relevant carrying out the analysis. It should define the purpose of the work and its significance. In this perspective, this section is too succinct and fails to effectively point out the relevance of your contribution towards the existing literature. 

Overall I suggest the authors to improve this section by discussing also about the notion of circular economy. Both product quality and service quality have always been an important matter for the sustainable development of enterprises and this is particularly relevant in circular economy model of business and stakeholders involvment.

Some literature to look at:

https://www.mdpi.com/2076-0760/8/7/216

https://www.sciencedirect.com/science/article/pii/S0921344919304598

https://www.sciencedirect.com/science/article/pii/S0959652620304224

https://www.sciencedirect.com/science/article/pii/S2351978919305372

https://onlinelibrary.wiley.com/doi/full/10.1002/csr.1791

Materials and methods: I found this section very important for the readability of the paper. However, several challenges need to be addressed. Methods should be described in detail. I think the research procedure could be much more clearly described by means of a diagram also highlighting its potential and limit.

Discussions: It is missing. I recommend to rearrange the sections of the paper following the journal guidelines for authors.

Conclusions: Conclusions must also be revised according to the previous comments. In particular, they should discuss practical and policy implications as well as future lines of research. As it stands now, they fail to extract all the juice of your work. 

Please adapt references as requested by the journal format

I hope these comments might help in improving the paper and encourage the authors to move forward.

Author Response

Dear Reviewer:

Thank you for your letter and for the reviewers’comments concerning our manuscript entitled Research on Quality Decisions and Coordination with Reference Effect in Dual-channel Supply Chain(ID: 731998). Those comments are all valuable and very helpful for revising and improving our paper, as well as the important guiding significance to our researches. We have studied comments carefully and have made correction which we hope meet with approval. Revised portion are marked in red in the paper. The main corrections in the paper and the responds to the reviewer’s comments are as flowing:

1.Response to comment:Abstract has inappropriate structure. I suggest to answer the following aspects: - general context - novelty of the work - methodology used (describe briefly the main methods or treatments applied) - main results and related interpretations.

Response: Thank you for pointing this out. According to you advice, we added this point into our revised manuscript and the details can be found in Line 14-32, Page 1.

2.Response to comment: Introduction: This section should briefly place the study in a wide context and emphasize why it is relevant carrying out the analysis. It should define the purpose of the work and its significance. In this perspective, this section is too succinct and fails to effectively point out the relevance of your contribution towards the existing literature. 

Response: Thanks for the referee’s kind suggestion. We added the purpose and significance of this study into our revised manuscript. The detailed revisions can be found in Line 101-107, Page ,and Line 88-91, Page 3. 

3.Response to comment: Overall I suggest the authors to improve this section by discussing also about the notion of circular economy. Both product quality and service quality have always been an important matter for the sustainable development of enterprises and this is particularly relevant in circular economy model of business and stakeholders involvment.

Response: Thanks for the referee’s suggestion. According to you advices, we first discuss the concept of circular economy and the goal of circular economy. Then it analyzes the relationship between quality and circular economy. Through three practical cases to illustrate the impact of quality on business operation. Finally, the importance of quality to the sustainable development of enterprises is expounded. The detailed revision can be found in Line 37-64, Page 1- 2. 

4.Response to comment:

Some literature to look at:https://www.mdpi.com/2076-0760/8/7/216

https://www.sciencedirect.com/science/article/pii/S0921344919304598

https://www.sciencedirect.com/science/article/pii/S0959652620304224

https://www.sciencedirect.com/science/article/pii/S2351978919305372

https://onlinelibrary.wiley.com/doi/full/10.1002/csr.1791

Response: Thanks for the referee’s kind advice. We added appropriate references from the list above.The detailed revision can be found in Line 733-738, Page 20. 

5.Response to comment: Materials and methods: I found this section very important for the readability of the paper. However, several challenges need to be addressed. Methods should be described in detail. I think the research procedure could be much more clearly described by means of a diagram also highlighting its potential and limit.

Response: Thanks for the referee’s kind suggestion.We added a Table and a Figure. The detailed revisions can be found in Line 232-233;304-309;336-339;372-373.We also added some contents to readability of the paper. The detailed revisions can be found in Line 270-273;246-266;et al..

6.Response to comment: Discussions: It is missing. I recommend to rearrange the sections of the paper following the journal guidelines for authors.

Response: Thanks for the referee’s kind advice. We take the discussion and conclusion as a section, which is conducive to the analysis of this paper. There are many added contents in this section. The detailed revision can be found in Line 552-633,Page 15-17. We rearrange the sections of the paper following the journal guidelines for authors. The proofs of Propositions are put at the back of the text.

7.Response to comment: Conclusions must also be revised according to the previous comments. In particular, they should discuss practical and policy implications as well as future lines of research. As it stands now, they fail to extract all the juice of your work. 

Response: Thank you for pointing this out. Your suggestion is very correct. We added practical and policy implications as well as future lines of research. The detailed revisions can be found in Line 552-643,Page 15-17.

8.Response to comment: Please adapt references as requested by the journal format.

Response: Thank you for pointing this out. According to the requirements of this journal, the format of references is modified.The detailed revisions can be found in Line 733-810,Page 20-21.

Reviewer 2 Report

Major comments:

The reviewed paper proposes an interesting problem, which is not only theoretically, but could be also practically important.

Author(s) focused on dual channel, quality, reference effect, differential game, and coordination.

Most important findings presented by author(s) are:

(1) The optimal quality under centralized decision-making is respectively greater than that under decentralized decision-making. When the relationship between members marginal profits meet a certain condition under decentralized decision-making with the cost-sharing contract, the retailer can actively improve service quality while the manufacturer do not.

(2) For the profits, when the relationship between members marginal profits meet a certain condition, the profits of members under decentralized decision-making with the cost-sharing contract are greater than that under decentralized decision-making with the no cost-sharing contract. In addition, the profit of supply chain under centralized decision is greater than that under decentralized decision.

(3) Under the decentralized decision-making, the service cost-sharing proportion is positively related to the manufacturer's marginal profits in traditional channel and online channel. The higher the manufacturer's marginal profits are, the more service cost will be borne by the manufacturer, so that the retailer will further increase the power of service cost investment, and then manufacturer's profit will increase. On the contrary, the cost-sharing proportion is negatively related to the retailer's profit margin. The higher the retailer's marginal profit is, the more active the retailer is in improving the service quality. Meanwhile, the manufacturer will bear more service cost, which affects the manufacturer's enthusiasm to share the retailer's service cost to some extent.

(4) For the service quality spillover effect, the monotonicity of retailer's profit changing with spillover effect of service quality has nothing to do with the initial reference service quality. Meanwhile, only under the coordination mechanism, the monotonicity of manufacturer's profit changing with spillover effect of service quality has nothing to do with the initial reference service quality. In addition, under decentralized decision-making scenario, when the initial reference service quality is relatively high, the higher the service quality spillover effect is, the worse the manufacturer's profit is.

(5) Under the coordination mechanism scenario, when the cost-sharing proportion meets certain conditions, it can make the supply chain system reach a coordinated state, in which the profit of the supply chain under the decentralized decision-making scenario is equal to the profit of the supply chain under the centralized decision-making scenario. Then the profits of both manufacturer and retailer can be improved.

(6) The reference quality increases with time and tends to be stable. Different initial reference quality will affect the paths of reference quality changing with time, but will not affect the final stable value. In addition, it is found that stability value of the reference quality under the coordination mechanism is greater than that under the decentralized decision-making.

Author(s) formulated a dynamic models that include the product quality reference effect and the service quality reference effect in a dual channel supply chain system consisting of a manufacturer and a retailer. The paper develops decision models with quality reference effect under the four different scenarios(decentralized decision making without cost-sharing contract, decentralized decision making with cost-sharing contract, centralized decision making, coordinating mechanism).

The paper has a logical structure and is clearly, concisely and accurately written.

I suggest to update abstract to highlight most important findings of this research.

“Introduction“ and “Literature review“ parts should be updated, the author(s) did not clearly show the difference between their approach and those in the literature. Complex and expanded state-of-the-art is needed.

I also suggest author(s) should add discussion about pros and cons of considered problem to clearly identify the benefits.

It would be interesting for readers if the paper include theoretical section about extended number of different potential practical applications in different areas, if possible. I suggest to add it.

The findings are too much dependent on the used approach and the case study to be generalized.

I suggest to add all appropriate references from the list below:

10.1016/j.jclepro.2019.119521

10.1016/j.jclepro.2019.119273

10.3390/app9173509

10.1016/j.ijpe.2017.06.031

10.1111/itor.12440

10.1016/j.trpro.2017.12.056

10.1016/j.ejor.2018.05.067

“Decentralized decision making without cost-sharing contract“, “decentralized decision making with cost-sharing contract“, “centralized decision making“, “coordinating mechanism“, mentioned by author(s), as presented are not clearly explained in the paper.

I suggest author(s) clearly explain motivation for considering these “four different scenarios”.

Minor comments:

Paper contains some amount of typos that need to be corrected throughout the paper. There are several minor language errors in the text. Some sentences require rewriting. Some acronyms were not defined.

Author Response

Dear Reviewer:

Thank you for the reviewers’comments concerning our manuscript entitled Research on Quality Decisions and Coordination with Reference Effect in Dual-channel Supply Chain(ID: 731998). Those comments are all valuable and very helpful for revising and improving our paper, as well as the important guiding significance to our researches. We have studied comments carefully and have made correction which we hope meet with approval. Revised portion are marked in red in the paper. The main corrections in the paper and the responds to the reviewer’s comments are as flowing:
Responds to the reviewers comments:

1.Response to comment:I suggest to update abstract to highlight most important findings of this research.

Response: Thank you for pointing this out. According to you advice, we added this point into our revised manuscript and the details can be found in Line 25-32, Page 1.

2.Response to comment:“Introduction“ and “Literature review“ parts should be updated, the author(s) did not clearly show the difference between their approach and those in the literature. Complex and expanded state-of-the-art is needed.

Response: Thanks for the referee’s kind suggestions. “Introduction“ and “Literature review“ parts was updated. The paper clearly show the difference between their approach and those in the literature, the details can be found in Line 180-183, Page 4. In the section of Introduction and Literature review, we added complex and expanded state-of-the-art.

3.Response to comment: I also suggest author(s) should add discussion about pros and cons of considered problem to clearly identify the benefits.

Response: Thanks for the referee’s suggestion. According to you advices, we added discussions about pros and cons of considered problems.The detailed revisions can be found in Line 565, Page 16; Line 574-576, Page 16; Line 583-584,Page 16;Line 596-598 Page 16; Line 590-591, Page 16; Line 602-605,Page 16; Line 626-630,Page 17.

4.Response to comment: It would be interesting for readers if the paper include theoretical section about extended number of different potential practical applications in different areas, if possible. I suggest to add it.

Response: Thanks for the referee’s kind advice. We added practical applications.The detailed revisions can be found in Line565-569, Page 16. 

5.Response to comment: The findings are too much dependent on the used approach and the case study to be generalized.

Thank you for pointing this out. Your suggestion is very correct. We added practical and policy implications as well as future lines of research.The detailed revisions can be found in Line 552-643,Page 15-17.

6.Response to comment: I suggest to add all appropriate references from the list below:

10.1016/j.jclepro.2019.119521

10.1016/j.jclepro.2019.119273

10.3390/app9173509

10.1016/j.ijpe.2017.06.031

10.1111/itor.12440

10.1016/j.trpro.2017.12.056

10.1016/j.ejor.2018.05.067

Response: Thanks for the referee’s kind advice. We added appropriate references from the list above.The detailed revisions can be found in Line 770-780, Page 20-21. 

7.Response to comment: “Decentralized decision making without cost-sharing contract“, “decentralized decision making with cost-sharing contract“, “centralized decision making“, “coordinating mechanism“, mentioned by author(s), as presented are not clearly explained in the paper.

I suggest author(s) clearly explain motivation for considering these “four different scenarios”.

Response: Thank you for pointing this out. we explained motivation for considering these “four different scenarios”. The detailed revisions can be found in Line 246-266, Page 6-7;Line 270-273.

8.Response to comment: Paper contains some amount of typos that need to be corrected throughout the paper. There are several minor language errors in the text. Some sentences require rewriting. Some acronyms were not defined.

Response: Thanks for your suggestions. Amount of typos and minor language errors were corrected. Some acronyms were defined.The detailed revisions can be found in Line 19;64-67;71-72;96;172;196;210;214-216;239-240;336-339;353;397-400;403-404.

Round 2

Reviewer 1 Report

Dear authors,

your revised version is very improved. Congratulations. 

Reviewer 2 Report

Accept